# Students' and staffs' views and experiences of asymptomatic testing on a university campus during the COVID-19 pandemic in Scotland: a mixed methods study

Linda Bauld ,[1] Alice Street ,[2] Roxanne Connelly,[2] Imogen Bevan ,[2] Yazmin Morlet Corti ,[2] Mats Stage Baxter,[1] Helen R Stagg ,[1] Sarah Christison ,[2] Tamara Mulherin,[1] Lesley Sinclair,[3] Tim Aitman [4]

¹Usher Institute, The University of Edinburgh, Edinburgh, UK
²School of Social and Political Science, The University of Edinburgh, Edinburgh, UK
³Department of Health Sciences, University of York, York, UK
⁴Institute of Genetics and Cancer, The University of Edinburgh, Edinburgh, UK

**Correspondence to**
Professor Linda Bauld;
Linda.Bauld@ed.ac.uk

## ABSTRACT

**Objectives** To explore the acceptability of regular asymptomatic testing for SARS-CoV-2 on a university campus using saliva sampling for PCR analysis and the barriers and facilitators to participation.

**Design** Cross-sectional surveys and qualitative semistructured interviews.

**Setting** Edinburgh, Scotland.

**Participants** University staff and students who had registered for the testing programme (TestEd) and provided at least one sample.

**Results** 522 participants completed a pilot survey in April 2021 and 1750 completed the main survey (November 2021). 48 staff and students who consented to be contacted for interview took part in the qualitative research. Participants were positive about their experience with TestEd with 94% describing it as 'excellent' or 'good'. Facilitators to participation included multiple testing sites on campus, ease of providing saliva samples compared with nasopharyngeal swabs, perceived accuracy compared with lateral flow devices (LFDs) and reassurance of test availability while working or studying on campus. Barriers included concerns about privacy while testing, time to and methods of receiving results compared with LFDs and concerns about insufficient uptake in the university community. There was little evidence that the availability of testing on campus changed the behaviour of participants during a period when COVID-19 restrictions were in place.

**Conclusions** The provision of free asymptomatic testing for COVID-19 on a university campus was welcomed by participants and the use of saliva-based PCR testing was regarded as more comfortable and accurate than LFDs. Convenience is a key facilitator of participation in regular asymptomatic testing programmes. Availability of testing did not appear to undermine engagement with public health guidelines.

## STRENGTHS AND LIMITATIONS OF THIS STUDY

⇒ Mixed methods study to explore perceptions of a novel saliva-based PCR asymptomatic testing programme for COVID-19 designed to improve on lateral flow devices in a screening context.
⇒ Included two surveys 6 months apart and in-depth semistructured interviews with a subsample of participants.
⇒ Limited to the views and experiences of those who chose to take part and could not explore reasons for non-participation or compare the characteristics of participants with the university population as a whole.
⇒ Findings may be transferable to other asymptomatic testing programmes for SARS-CoV-2 or other viruses on university campuses or in other educational settings and workplaces.

## INTRODUCTION

The extent to which universities played a role in community transmission of SARS-CoV-2 was heavily debated in the UK in the early stages of the COVID-19 pandemic.[1 2] As in many other countries, higher and further education institutions had to pause non-essential teaching and research activities on several occasions, leading to long periods of online learning and many staff working from home. Essential campus activities continued throughout, however, and students moved between their term-time accommodation and other locations. In order to improve the safety of on-campus activities and reduce the risks of outbreaks, some experts recommended regular asymptomatic testing of students and staff alongside other public health measures.[3–5]

A few UK universities were early adopters of this approach, establishing their own pilot asymptomatic testing programmes involving either PCR[6 7] or lateral flow devices (LFDs).[8 9] Early studies of these programmes found acceptability of asymptomatic testing

among students to be high. However, uptake and adherence were found to be affected by anxiety[6] and concerns about the accuracy of tests, especially LFDs,[8 9] raising questions about students' long-term willingness to engage with regular testing. Government-funded asymptomatic COVID-19 testing in the form of LFDs first became available to all UK universities in December 2020 following concerns that a mass 'migration' of students over the winter break might lead to a rapid rise in cases.[10] This was offered to all students leaving and returning to campus. Evaluations of this testing programme found uptake among students to be low[11] and concerns about accuracy were a prominent barrier to participation.[11 12] LFDs were rolled out to the general public from April 2021. Students and staff were then encouraged to test two times per week using LFDs. However, given their low sensitivity, several experts have queried the benefits and cost-effectiveness of mass asymptomatic LFD testing, especially during periods of lower viral prevalence in the community.[13–15]

The University of Edinburgh established an asymptomatic testing research programme, TestEd (www.ed.ac.uk/tested-covid), in January 2021. This aimed to improve on existing approaches to PCR testing in terms of acceptability and cost, and also provide a more accurate alternative to LFDs. TestEd involves a novel testing platform that uses pooled saliva-based testing by PCR, with a protocol adapted from an approach for nasopharyngeal swab testing.[16]

TestEd included surveys and interviews with participating staff and students to explore: the acceptability of regular PCR testing among students and staff, particularly involving an approach that was less invasive than nasopharyngeal swabbing; barriers and facilitators to participating in a regular university testing programme, including in the context of other testing methods being available; and whether participation in such a programme changed adherence to public health guidelines. We suggest that understanding staff and students' perceptions and experiences of TestEd's novel testing system can help to inform the design of effective regular asymptomatic testing programmes for COVID-19 or other disease outbreaks in educational and workplace settings in the future.

## METHODS
### The TestEd programme
All University of Edinburgh students and staff who were coming onto campus were eligible to take part in TestEd on a voluntary basis and could sign up at any time. After joining, they were encouraged to provide two times per week saliva samples at one of the thirty testing centres located throughout the university. This involved spitting into a plastic cup, transferring the saliva to a tube and scanning their participant identifier and a barcode on the tube to register their sample. Samples were then collected from test sites and transferred to a university lab for PCR analysis. Participants normally receive their test results within 24 hours by logging onto a secure portal with their

university username and password. Between January 2021 and February 2022, 3895 staff and 3106 students registered and consented to participate. The programme tested just over 100 000 samples with more than 170 positive results during that period. A supermarket shopping voucher was provided to those who tested positive and sought a confirmatory PCR test from the National Health Service (NHS) to assist with self-isolation. Participants were asked not to travel to campus to access testing, but instead to use TestEd while already there to study/work.

### Design
Participants who consented to taking part in TestEd and who had provided at least one saliva sample were invited by email to participate in two online surveys using the Qualtrics tool, one (a pilot) carried out between 15 April and 30 April 2021 and the main survey between 8 November and 21 November 2021. The pilot and main surveys consisted of closed-ended and open-ended questions (see online supplemental files 1 and 2[17]). The pilot survey was tested with three post-graduate students and amended following their feedback prior to distribution. No questions were compulsory. The number of eligible TestEd participants increased between the pilot and the main survey when students and more staff returned to campus for the 2021/2022 academic year.

Semistructured online interviews with participants were conducted between May 2021 and February 2022 (see online supplemental file 3). We were particularly interested in the views of participants who tested positive and aimed to interview more of this group than those who tested negative. We used purposive sampling to recruit participants from across the university and a wide range of demographic groups (university role, age, gender, ethnicity and disability) in order to ensure a diverse range of views and experiences were represented.

### Patient and public involvement
Volunteer students and staff were involved in contributing to the survey design and testing the questionnaire before the survey launched.

### Analysis
Data from both surveys were extracted from Qualtrics and exported to Stata. Variable recoding was undertaken to enable appropriate cell sizes for statistical analysis and to avoid statistical disclosure (>15). Variables indicating gender, age, ethnicity, disability and university role were recorded. Recoding as missing values was applied for all variables with 'not applicable' and 'prefer not to say' responses. Due to small numbers, the 'non-binary' category of gender was recoded as missing and the categories of ethnicity were grouped as shown in table 1. Responses to the survey questions were examined using descriptive statistics (eg, frequencies and percentages). We conducted $\chi^2$ tests and Fisher's exact tests where appropriate in order to investigate patterns between sociodemographic characteristics and responses to the survey. While

**Table 1** Participant characteristics

| Participant characteristics*† | Main survey n (%) | Pilot n (%) | Interviews |
|---|---|---|---|
| **Overall** | **1750** | **300** | **48** |
| Gender | | | |
| Female | 996 (58%) | 194 (65%) | 26 |
| Male | 721 (41%) | 103 (34%) | 21 |
| Non-binary | 21 (1%) | 2 (<1%) | 1 |
| Other | 1 (<1%) | 0 (<1%) | 0 |
| Preferred not to disclose | 33 (<1%) | 3 (<1%) | 0 |
| Age (years) | | | |
| ≤19 | 41 (2%) | 0 (0%) | 4 |
| 20–29 | 512 (29%) | 77 (26%) | 13 |
| 30–39 | 403 (23%) | 73 (24%) | 11 |
| 40–49 | 336 (19%) | 54 (18%) | 9 |
| 50–59 | 335 (19%) | 60 (20%) | 5 |
| ≥60 | 123 (7%) | 36 (12%) | 6 |
| Ethnicity | | | |
| British/Irish/other white | 1570 (90%) | 272 (92%) | 33 |
| Asian/Indian/Pakistani/Bangladeshi/Chinese/other Asian | 98 (6%) | 13 (4%) | 9 |
| Mixed/other ethnic/other black/Caribbean African | 71 (4%) | 11 (4%) | 6 |
| Preferred not to disclose | 11 (<1%) | 1 (<1%) | 0 |
| Role in the university | | | |
| Staff | 1247 (72%) | 248 (83%) | 28 |
| Student | 482 (28%) | 52 (17%) | 20 |
| Preferred not to disclose | 21 (<1%) | 0(<1%) | 0 |
| Disability | | | |
| Yes | 46 (3%) | 6 (2%) | 5 |
| No | 1651 (97%) | 284 (98%) | 40 |
| Preferred not to disclose | 53(<1%) | 10 (<1%) | 3 |

*Original values are retained; the analysis groups responses <15 into categories.
†Sociodemographic characteristics were collected at TestEd registration.

some of these tests were statistically significant, effect sizes were very low (Cramér's≤0.1) indicating only very weak patterns of association. These results are not presented in the main text and are available instead in online supplemental file 4. For questions that were duplicated in the two surveys, where participants had responded to these both times, it was possible to analyse changes in attitudes and experiences between the two time points.

Qualitative data from open-ended survey questions and semistructured interviews were analysed through a thematic coding approach by SC, IB and AS using NVivo software (V.1.3 and V.1.6.1). The content of the survey questions provided an initial coding structure, which was revised during analysis to reflect additional issues and topics raised in the results. Coding of semistructured interviews was inductive, reflecting the more open-ended nature of the interviews. The interviews addressed a wide range of topics and for this article we only analysed a subset of results related to acceptability, perceptions and experiences of the TestEd programme. Initial coding was carried out by SC (survey) and IB (interviews) and quality checked by AS who read all results and interview transcripts. Coding categories were collectively reviewed, discussed and revised as a team before a final coding structure was agreed for each dataset. Codes were collectively organised into themes by SC, IB and AS during team analysis meetings. The team discussed and analysed commonalities, overlaps and differences between the codes to derive common themes. A shared table on Microsoft Sharepoint was used to visualise relationships between example data extracts, codes and themes to ensure that clear connections could be drawn between the analysis and the data and to check where thematic categories were too narrow or broad. Themes were collectively reviewed against a sample of the surveys to check for coherence and areas of overlap between themes, with iterative changes

made to the thematic scheme. The agreed set of themes are reflected in the subheadings below.

## RESULTS

Out of 760 eligible participants who had provided at least one saliva sample when the pilot survey was distributed, 548 responses were received (72%), 522 of which were complete (69%). For the main survey, out of 4512 eligible participants, 2995 responses were received (66%), 1750 of which were complete (58%). A total of 300 participants responded to both surveys. A total of 70 participants were invited for interview, 48 of whom were successfully contacted and took part.

### Participant characteristics

Participant characteristics are shown in table 1. When compared with TestEd participants overall, the survey population included more staff members and participants identifying as female (data not shown).

### Reasons for participating

Overall, survey participants were positive about their experience with TestEd. 74% rated the experience as 'excellent' and 24% as 'good' in the main survey (see online supplemental file 4). Those who participated in both the pilot and main surveys maintained enthusiasm for the programme over time, with little change in responses.

Survey responses indicated that 'knowing (their own) COVID-19 status in the absence of symptoms' was the most important reason for participation (38%), followed by prevention of 'passing on infection to family and friends' (32%). For 18% of respondents, the most important reason was 'to contribute to scientific research on COVID-19', and for 11% this was 'to prevent passing on infection to other colleagues/students on campus if I am positive'.

Interview participants similarly emphasised their desire to protect family and friends beyond the university community as being a primary motivation for joining the programme. While knowing their own COVID-19 status was considered important, this was often linked to the benefit of protecting others inside or outside the university, rather than viewing these as separate benefits of testing. Some interview participants described previous negative personal experiences of COVID-19 or their witnessing of COVID-19 or Long COVID symptoms in friends and family as a motivation to test, to prevent passing on the infection to others. Some interview participants also emphasised the heightened need for testing post vaccination, when symptoms might be mitigated but one might still be infectious to others. The rationale of contributing to scientific research often emerged as an additional but secondary concern for interviewees. Other factors that interviewees suggested motivated them to join TestEd included the perceived need to follow government or institutional guidance; support and encouragement from the institution to take part; influence from peers; perceptions of risk; and, in a few select cases, the experience of COVID-19-like symptoms.

### Testing method

Survey participants found the simple spit test easy to administer and less invasive compared with standard PCR or LFD swab-based tests. They found the process of providing a saliva sample to be fairly quick: for 42% of respondents, it took only 2–5 min out of their day; 41% indicated that it took just 1–2 min.

However, the saliva testing was not without problems. A few participants indicated that it could be difficult to produce enough saliva to provide a viable sample. This was also raised in interviews. Staff and students who signed up to TestEd were asked not to eat or drink for 30 min before testing. Some survey participants described this as a limitation, indicating that they would find it more convenient to provide a sample during their lunch or coffee breaks. There were also some issues with the privacy of sample collection booths, with some people feeling uncomfortable spitting into a cup when they could be observed. The booths did have sides but were located

| Table 2 Views on the TestEd testing method | |
|---|---|
| **Facilitators** | **Barriers** |
| It's non-invasive, simple, and involves no discomfort whatsoever. This is a huge benefit in making a testing regime attractive to its users. | If the spit sample is not of a high enough volume it will not work. So sometimes my results may have been invalid. I have to work up spit in my mouth for a couple of minutes prior. |
| A much less invasive form of testing compared to lateral flow tests! Given how invasive they are, I also doubt many are correctly using other lateral flow tests, rendering the results inaccurate. | Sample can be given easily on the way to school. The only inconvenience comes from the time taken to collect enough saliva for the sample and finding a time where I have not eaten or drank in the past 30 minutes. |
| Saliva samples are very easy to provide (and non-intrusive) and I was concerned that I may not have been doing the lateral flow nasal and throat swab correctly hence my preference for saliva sampling. | I'd prefer a privacy curtain that I could pull behind me when I'm in the booth. I feel very exposed when spitting in the cup in the middle of the library, especially if things get messy! |
| It is very convenient and much more accessible than doing a tonsil/nostril swab. Saliva spit tests increase my motivation to test. | I felt very much under pressure to do this spitting thing, and I couldn't perform basically, so I just took everything with me in the office and I was like, "I'm nice and safe here." There was nobody around, but still it felt very weird to have to spit. |

in public venues on campus. Table 2 reports a selection of participant views on the testing method.

## Convenience

The majority of survey participants also indicated that it was either 'very convenient' (68%) or 'convenient' (26%) to provide a sample as part of their work/study schedules (see online supplemental file 4). Participants touched on issues of convenience at multiple points in the testing process, from experiences of sample collection, to navigating the TestEd IT systems, to the receipt of results.

Participating in TestEd was reported to be convenient due to the number and location of the test centres, which were in many cases located within buildings where participants worked. Participants also described how the drop-in element made participating easier as tests could be taken at any time without appointment or prior booking. Interview responses revealed that perceptions of convenience were often linked to individual work/study patterns. Interviewees with a regular on-campus working schedule, and particularly those with a testing site inside the same building as their office or laboratory, reported developing a routine testing schedule (eg, on the way to work or during a morning break) by comparison with students and staff with more varying schedules and who worked across multiple locations or between home and campus, who found it harder to establish a testing routine.

Some expressed a desire for longer opening hours, better communication of opening hours, or complained that some centres were not open as advertised or could not be accessed without the correct security clearance on their staff/student card. Many participants also found the testing programme's IT systems cumbersome. In a few cases, people reported that the bar code on the test tube did not work. Many participants described the process for logging in and accessing results through the online system to be inconvenient and expressed a preference for the NHS system of sending results directly via SMS and email. While some found the turnaround times to be 'quick', other participants described turnaround times to be inconsistent or too long. Further probing in interviews revealed that perceptions of test turnaround times as either quick or slow were often shaped by comparison with another form of testing (eg, LFD, NHS-administered PCR test), and by specific time-sensitive motivations for testing on that occasion. Table 3 presents some of these views about perceived convenience.

## Concerns about COVID-19 on campus

Most survey participants indicated some level of concern about catching COVID-19 on campus: 21% were 'very concerned'; 33% were 'moderately concerned'; and 22% 'somewhat concerned'. Many expressed concerns about the return of students and the reintroduction of in-person teaching, which were perceived to have led to increased mixing on campus. A common concern was the lack of adequate ventilation in teaching rooms and the ability to maintain social distancing in shared spaces:

| Table 3 | Convenience of TestEd |
|---|---|
| **Facilitators** | **Barriers** |
| 'There is a testing station at my university accommodation so it is very easy to get to and provide a sample.' 'The booths are close to my work area. The process is quick, so you can easily fit in your schedule. Also it's self-administered and open all the time, so you can test anytime.' 'Station is set up throughout working hours, drop-in nature means can give a sample at a time that suits in my clinical day.' | 'I'm either out on site so I'm at(campus site 1), or I'm at(site 2)– it then becomes a question of, "Do I have the time to drive from those locations back to(site 3)for a ten-minute spit test?" So at times you just have to sacrifice the test and not go.' 'It seems unnecessary to have to log in to get my results once notified. The NHS system doesn't require this: the text message and email both contain the test result.' 'Sometimes the results take longer to come through than other times – it can be hard to know how long to expect to wait for results.' |

I slightly worry that I may catch Covid-19 from a student in class, as I spend a good amount of time with my students and not all our rooms are as well ventilated as I'd like them to be.

There is obviously some increased risk due to meeting more people and using more shared facilities than if working at home.

Despite these concerns, many survey participants perceived the likelihood of infection on campus to be lower than elsewhere. While some felt that there was low compliance with safety measures, others believed that the university's infection control measures were robust and effective. Some of these different perspectives of safety on campus may be related to a participant's position or role within the university—for example, working alone in single-occupancy offices versus roles that involved more contact with others at work or while studying:

I felt that the safety precautions in operation at work (mask-wearing, handwashing, social distancing) were adequate.

[I am] usually based in my office which is single occupancy – risk here is less than going to the shops.

## Reassurance

The majority of survey participants (87%) indicated that the availability of the TestEd programme made them feel reassured about working or studying on campus. Levels of reassurance increased over time among participants who took part in both the pilot and main surveys, rising from 90% to 94%.

In some cases, TestEd provided reassurance about participants' own health, but it was more common for participants to connect that reassurance to their sense of personal responsibility for the well-being of others.

**Table 4** Reassurance

| Facilitators | Barriers |
| --- | --- |
| (I)am severely immunocompromised so worried about all contact. Knowing the majority are being tested regularly has eased these concerns.<br><br>I don't think it makes a significant difference to my risk of catching Covid on campus, but it reduces the risk that I might unknowingly pass on Covid.<br><br>It's good that my colleagues and I have access to a free and accurate testing service, so I am confident that I am not unwittingly spreading COVID.<br><br>Most of my direct colleagues are using TestEd as well. Reassuring when working in the same room.<br><br>Knowing that colleagues were also participating in the programme provided a certain level of reassurance, along with my own results of course.<br><br>Because as the staffing levels have increased, I see an increasing amount of provided samples in the collection trays so I am confident people are getting regularly tested. | It's unclear to me how many staff and students are taking part and how regular(ly) they are testing, so it doesn't necessarily make me feel more reassured about catching COVID-19 while at work.<br><br>On one hand it is definitely a positive, but on the other I often see how few samples have been submitted when I go to drop off my own. It doesn't seem like as many people have taken advantage of the availability of the system as could have.<br><br>There seems to be very little take up on it – maybe only 15–20 samples when I go so (I am) concerned a lot of people, especially students, aren't doing it. I'm aware of outbreaks on campus but we're not officially informed of that – I think we should be.<br><br>If everyone on campus was required to enrol in TestEd to work/study on campus, I would feel safer. Voluntary enrolment is not good enough to ensure safety.'<br><br>'I'd feel more assured if it was compulsory for all who use campus. Some of my students think they are immune and are less risk-averse as a result. |

Responses to open-ended survey questions and interviews indicated that perceived levels of participation among others influenced how reassured individuals felt. Those who felt that there were high levels of participation, particularly among close colleagues, indicated that this made them feel reassured, while those who thought those around them on campus were not taking part had more concerns. One factor that influenced how participants perceived participation was the number of samples which they noticed had been provided at test sites. Concerns about low participation led some participants and interviewees to reflect on the efficacy of a workplace testing programme that relied on voluntary participation. Table 4 highlights a number of these responses.

### Accuracy

The vast majority of survey participants (92%) agreed that the results they received from TestEd were accurate. Among those who completed both the pilot and the main survey, 93% indicated in both surveys that they believed their results from TestEd were accurate.

In open-ended survey responses and interviews, participants emphasised their trust in the scientists involved in developing the TestEd programme as a basis for their belief in the accuracy of the test results. Participants also described how they had more faith in the PCR testing used for TestEd compared with LFDs. Some also reported that they felt the saliva-based tests were likely to be more accurate, as the sample collection process was less prone to user error compared with self-administered swabs.

While participating in TestEd, many people were also using other testing methods, most commonly LFDs that were freely available in a variety of venues, including on campus. In the case of a positive TestEd result, all participants interviewed carried out a confirmatory PCR test through the NHS so that a positive test picked up in the study could be formally reported, allowing for contact tracing by the NHS. Testing positive via this confirmatory NHS test also confirmed for many that TestEd's methods were accurate. Interviewees also reported using LFDs either to confirm a positive TestEd result or to check the accuracy of LFDs compared with PCR.

For the small proportion of participants who were unsure about the accuracy of TestEd results, open-ended responses indicated that more information regarding the effectiveness of saliva-based testing could provide reassurance. Some of this concern over accuracy was linked to the novel nature of the approach, with several participants stating that they felt there was a lack of knowledge regarding the effectiveness of saliva-based testing or that the programme was an experimental study to trial this type of testing methodology (see table 5).

### Compliance with public health guidelines

Respondents were asked whether they had changed their approach to the public health guidelines that were in place at the time of the study (ie, social distancing and face coverings) at work or study since they joined TestEd. The majority (93%) indicated that they had not changed their approach. Only 5% reported that they had and 2% did not know. Responses to this question were similar between the pilot and main survey.

Among the small number of participants who indicated that they had changed their behaviour, some participants reported feeling more relaxed with regard to guidelines. In some cases, this made them less adherent and in other cases it made them more confident to mix with others within the guidelines. Others who reported changing their behaviour following participation in TestEd explained that the testing programme had resulted them in following guidelines more stringently, for example, with reference to wearing face masks:

> I was careful before as I wore FFP2 masks when in enclosed spaces. I am more reluctant to visit crowded public spaces as I worry that I could then test positive.

 Bauld L, *et al. BMJ Open* 2023;**13**:e065021. doi:10.1136/bmjopen-2022-065021

**Table 5** Accuracy

| Facilitators | Barriers |
|---|---|
| Because I trust the science behind it and I don't believe that it would have been rolled out university-wide if the university and the people behind TestEd were not confident that it would work.<br>I understand TestEd used a PCR test which the NHS says is more accurate than a lateral flow test.'<br>'The quality of the sample provided is independently verified by the TestEd research team. Providing a saliva sample is also more straightforward and likely more error-free.<br>I mean the PCR test from the NHS was positive as well so I'm pretty sure it [TestEd] was [accurate]. With not having any symptoms, and then I got a positive, I might have been, "Oh, I'm not 100% sure." But having both tests positive, I'm pretty sure it has been accurate. | I am unsure about the effectiveness of the saliva as compared to the nasal swab, and have not seen data to show that. I also don't know if there are therefore not a lot of false negatives.<br>Haven't heard of a positive result yet, I haven't seen any information of a direct comparison of this test and the [nasal] swab test so I would trust a swab test more.<br>PCR tests are the most reliable – although the saliva samples are obviously part of a trial so a bit of an unknown, but still feel confident it will pick up most positives, and probably more accurately than a lateral flow. |

Am less worried about interacting with friends and family given negative tests, so I see more people if I've been regularly testing.

I confess I am a little less strict than before in following the guidelines. I sometimes forgot I do not wear a mask. This may be due to the fact that I feel less worried about catching it.

In interviews, all participants who had tested positive reported having booked a confirmatory test through the NHS, to have informed their workplace and to have fully complied with self-isolation guidelines. However, some also indicated challenges, including the effects of their decision on others to self-isolate, financial consequences, impacts on personal well-being and a reliance on their own social networks for emotional support and provisions during the isolation period. Some interviewees, particularly students, highlighted issues such as taking out the rubbish, accessing meals and negotiating spaces with other members of a household who had not tested positive:

I was kind of really bored in my room, because in my flat there's one other person so I tried my best not to go in the kitchen or the living room. The only true place I can go is my bedroom and the bathroom. So it was quite difficult because I felt like I was also inconveniencing her; if I wanted water or food or something she had to bring it to my door. Although I am more than capable of making myself a cup of tea, I didn't want to go into the kitchen and accidentally contaminate things.

I didn't want to trouble other people to carry all my groceries for me. And it's not enough to stack up to the minimum delivery. So I just ended up trying to make do, asking people if they could just buy one or two things for me and stuff. So that's one very big inconvenience.

## DISCUSSION

This study adds to evidence from previous research that routine asymptomatic testing for SARS-CoV-2 can be introduced on university campuses in a way that is accessible and acceptable to staff and students. Although TestEd was used by a minority of students and university employees during the study period, the programme was introduced at a time when working from home guidance was in place and footfall on campus was low.[2] For those who regularly participated, enthusiasm for the availability of free asymptomatic testing was maintained over time.

Reasons for taking part included participants wanting to know their own COVID-19 status and avoiding passing the virus onto others, which confirms findings on attitudes to COVID-19 testing from studies in multiple countries.[18] Despite TestEd being a workplace programme, concern for others was not necessarily limited to colleagues and instead also related to protecting vulnerable friends and relatives off-campus. Early in the pandemic, it was suggested that highly interconnected social networks inside and outside university make it a high-risk environment.[19] Our findings suggest that university staff and students are aware of these risks and are willing to take active measures to reduce them.

Previous research has shown that concerns about physical discomfort and the capacity to perform nasopharyngeal swab-based sample collection are barriers to participation in testing.[20 21] TestEd involved a novel saliva-based sampling method for PCR testing, avoiding nasal pharyngeal swabs. Participants reported that this was a more comfortable form of testing. However, there were some concerns about producing enough saliva and around privacy while spitting into a cup at testing sites. Other university-based studies have found similar concerns among participants about their ability to perform saliva-based testing.[22 23] One study that compared saliva-based and swab-based testing methods found no consensus among participants on the preferred method.[6] While saliva-based testing has some advantages over swab testing in terms of physical comfort, our findings show that it can also introduce new challenges and concerns for participants.

The convenience of testing was something participants valued, confirming findings from other studies that have found convenience to be a key facilitator for COVID-19

testing uptake.[8 9 23 24] Aspects of the TestEd programme that were found to be convenient included the sample collection method and the quantity and accessibility of sample collection points across campus. However, in some instances, negative experiences of IT systems used to sign up, submit samples and access results negatively affected perceptions of convenience. Having to wait for results (compared with the quick turnaround time for LFDs) was also a disadvantage. Our findings show that the perceived convenience of a particular testing method varied in relation to the context for and purpose of testing. Because TestEd was in place when other forms of free testing were available (via the NHS for those with symptoms and LFDs for asymptomatic testing in wider society), it is unsurprising that participants combined different kinds of tests according to which was deemed most convenient at a particular moment.

Participation in TestEd was reported as being reassuring for participants, consistent with previous research on COVID-19 testing in education settings[6 8 23] and workplaces.[25 26] Our results found that this reassurance was, however, mediated by perceptions about levels of participation in the testing programme by others. Participants were sensitive to the question of whether they were part of a larger testing community, in part because they understood that the effectiveness of the programme as a public health screening tool depended on others also taking part.

Previous studies have found that concerns about the accuracy of LFDs can be a barrier to participation in testing.[7 9 11] We found that survey and interview participants were aware of differences in the sensitivity of PCR compared with LFDs, and perceived PCR to be a more accurate testing method. Saliva-based self-testing was also perceived to be more accurate than self-testing with a nasopharyngeal swab. Participating in a programme developed by university scientists provided some reassurance that testing results were likely to be accurate.

There was limited evidence that testing resulted in changes in behaviour among those who participated, for example, leading to increased confidence to socialise, both within and outside existing guidelines. We also did not find evidence that the availability of on-campus testing made participants more cautious or aware of COVID-19 guidelines, but it is likely that those engaging with TestEd were already aware of and trying to follow these guidelines. Similar findings have been reported for other university-based studies[6 8] and for workplace studies of antibody testing.[26] In line with findings from previous studies,[6 27 28] participants experienced daily challenges during self-isolation, such as when isolating from other members of the household,[29] but this did not affect self-reported compliance with guidelines.

We collected information about participant characteristics but did not identify any significant differences in survey responses between groups, although our samples may have been too small to examine relevant characteristics (such as disabilities or ethnicity) in detail.

An important limitation of this study is we could not assess what proportion of eligible students and staff accessed TestEd, because registration was intended for those who were coming onto campus, something that was not routinely monitored particularly as 'working from home' guidance varied at different stages of the pandemic and the study period. Many registered students and some staff worked entirely from home (including in other parts of the UK and overseas) throughout the period when the study was taking place.

In addition, the study did not include the views of staff or students who did not participate in TestEd, despite visiting campus during the study period. There were varying public health regulations and guidance in place over the period of the research[2] and limited available information about footfall on campus, given the size and complexity of a large university. It is therefore difficult to assess how many staff and students would have used the programme if everyone eligible to do so had signed up. In order to begin to understand reasons for non-participation in TestEd, we have recently engaged with the University of Edinburgh student panel, a group of 250 students designed to be representative of the student population. While almost all of those who responded to our brief online questionnaire to the panel (n=76, 30% response rate) had heard of TestEd, most chose not to participate because they did not get round to registering, preferred not to know if they had COVID-19, or used LFDs instead.

Engagement with TestEd is voluntary, meaning that the participant population may differ from the student and staff population as a whole. We could not explore further differences between the TestEd population and the university population due to a lack of available data. Survey participants may also differ from the wider population of TestEd programme participants. The survey response rates were reasonably high (72% for the pilot survey and 66% for the main survey). However, when comparing the characteristics of the survey respondents to all TestEd participants, we noted some differences, for example, that there were more women among the survey respondents. There may therefore be biases in the survey responses due to the nature of the survey sample.

## CONCLUSION

Despite alternative testing options being available in the community at the time of the research, our results indicate that an asymptomatic SARS-CoV-2 testing programme designed specifically for university staff and students was acceptable and was positively received by those who took part. The provision of multiple testing sites across campus and the ease of saliva sampling compared with swabs were facilitators to participation, as were perceptions about the accuracy of results from PCR testing compared with LFDs. Potential barriers to participation included concerns about privacy when providing a sample; difficulty in accessing and using IT systems; time to receiving results;

and concerns about the extent to which the testing would reduce the risk of outbreaks on campus in the case of low levels of participation in the programme. Perceptions of convenience shaped facilitators and barriers to participation at every stage of the testing process. The availability of testing did not appear to undermine protective behaviours among participants to follow COVID-19 guidelines. These findings suggest that saliva-based PCR asymptomatic testing offers an acceptable and alternative and/or complement to LFD asymptomatic testing on university campuses. Future studies should explore reasons for non-participation in testing programmes in similar workplace or educational settings.

**Acknowledgements** We would like to thank Professors Neil Turok and Nick Gilbert (TestEd coinvestigators) for their support for the survey and qualitative research elements. In addition, Professor Jonathan Seckl who facilitated University pump priming funding for setting up the programme and assisted with liaising with key University staff and student networks. Kathryn Carruthers was TestEd project manager when the study took place and her assistance was invaluable. Finally, we thank the University of Edinburgh staff and students who participated in the research.

**Contributors** LB, AS, HRS, RC and TA contributed to the design of this study. LB and TA initiated the project. LS and TM managed project the research. Statistical analysis was done by YMC and MSB supported by RC and HRS. Qualitative analysis was conducted by IB and SC supported by AS. All authors contributed to the manuscript and read and approved the final manuscript.

**Funding** This work was supported by UKRI Research Grant MR/W006243/1 and the University of Edinburgh. Additional support for manuscript preparation was from the DiaDev project funded by the European Research Council (ERC) European Union's Horizon 2020 Research and Innovation Programme, grant agreement 71540.

**Competing interests** LB is Chief Social Policy Adviser to the Scottish government (part-time secondment) and chaired the Universities and Colleges Advisory Group, a subgroup of the Chief Medical Officer of Scotland's Advisory Group on COVID-19. HRS has received payment from the Scottish Parliament for advising the COVID-19 recovery committee. AS receives funding from the European Research Council (grant number 715450) for Investigating the Design and Use of Diagnostic Devices in Global Health and holds positions on the Royal Anthropological Institute Medical Committee (unpaid) and the Wellcome Trust Career Development Committee (paid). TA receives internal support from the University of Edinburgh. As the founder and director of BioCaptiva (a liquid biopsy company unrelated to the present study), he receives consulting fees. Additionally, he has received travel expenses for the Biomarkers UK Congress, Oxford Global, November 2021, and Liquid Biopsies, Global Engage conference, December 2021. TA is the Regional Champion for Scotland for the Academy of Medical Sciences, and sits on the Genomics England Scientific Advisory Committee, European Research Council advanced grant panel for genetics. He is also a trustee and director of the PHG Foundation.

**Patient and public involvement** Patients and/or the public were involved in the design, or conduct, or reporting, or dissemination plans of this research. Refer to the Methods section for further details.

**Patient consent for publication** Not applicable.

**Ethics approval** This study involves human participants and was approved by the University of Edinburgh's Medical School Research Ethics Committee on 1 April 2021 Rec Ref: 20-EMREC-023_SA03. Participants gave informed consent to participate in the study before taking part.

**Provenance and peer review** Not commissioned; externally peer reviewed.

**Data availability statement** Data are available in a public, open access repository. LB, TA, HRS, RC, MSB, YMC, SC, AS. TestEd Survey of Staff and Student Experiences and Perceptions of Novel COVID-19 Testing Platform, 2021 (dataset). University of Edinburgh, Edinburgh Medical School, Usher Institute. https://doi.org/10.7488/ds/3802.

**ORCID iDs**
Linda Bauld http://orcid.org/0000-0001-7411-4260
Alice Street http://orcid.org/0000-0001-7874-0234
Imogen Bevan http://orcid.org/0000-0003-1835-4141
Yazmin Morlet Corti http://orcid.org/0000-0002-0168-3320
Helen R Stagg http://orcid.org/0000-0003-4022-3447
Sarah Christison http://orcid.org/0000-0002-5825-7277
Tim Aitman http://orcid.org/0000-0002-7875-4502

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
