## [Reviewer comments · BMJ Open]

ARTICLE DETAILS

TITLE (PROVISIONAL)	Student and staff views and experiences of asymptomatic testing on a university campus during the Covid-19 pandemic in Scotland: A mixed methods study
AUTHORS	Bauld, Linda; Street, Alice; Connelly, Roxanne; Bevan, Imogen; Morlet Corti, Yazmin; Baxter, Mats; Stagg, Helen R.; Christison, Sarah; Mulherin, Tamara; Sinclair, Lesley; Aitman, Tim

VERSION 1 – REVIEW

REVIEWER	Sarah Denford University of Exeter, Sport and Health Science
REVIEW RETURNED	19-Aug-2022

GENERAL COMMENTS	Many thanks for the opportunity to review this manuscript. The authors describe a well conducted mixed methods study exploring student and staff views and experiences of university testing using PCR tests for COVID-19. Of particular note is that the program involved the use of PCR tests (rather than LFDs), and this provides a novel perspective over and above many other university testing programs. Although the pandemic context has now changed considerably (most notably testing is no longer available/ required) this work still has considerable merit and importance for future outbreaks and/or pandemics. Introduction The introduction is clear, precise, and informative. The context is clearly described and the study is well justified. Methods Could the authors provide any details about what percentage of staff and students registered and consented to participate, and if known, what percentage this is of those who regularly attended campus? The authors note that there were 170 positive results during that period and that a shopping voucher was provided to positive cases in which a confirmatory PCR was sought. Were the 170 positive tests based purely on university PCR tests? Or is this those that were also confirmed through NHS PCR tests? Were you able to obtain data on the number of participants who also sought a PCR, and if so, how reliable were the university tests for detecting infection? The description of the qualitative analysis could be more detailed – for example, how did the authors move from codes to themes? Were any approaches used to enhance rigor/credibility of the analysis? Ethical approval should be mentioned. Results and discussion The results and discussion are informative and interesting. That testing did not (really) change behavior is an interesting finding. I would be interested to hear the authors' views about this, why this may have been (e.g., what did behavior before testing look like) and
--

	whether or not behavior (within the guidance) should have changed (which is arguably/debatably the point of testing).
--	---

REVIEWER	Mike Gill University of Surrey, Faculty of Health and Medical Sciences
REVIEW RETURNED	14-Dec-2022

GENERAL COMMENTS	The main limitation of this paper is the uncertainty about representativeness of the participants. They represent <15% of the University headcount of staff and students, and an unknown proportion of those actually attending the university during the study period, themselves not representative of all university staff and students, and varying in numbers over time. Both in the overall numbers who registered and consented to participate and in those who were surveyed, staff are overrepresented. Many of those who did participate did so for only limited periods: twice a week testing for all the 3895 staff and 3106 students for this length of time (say 53 weeks) would give rise to 742000 tests, rather than 'just over 100000'. Asking a small portion of an already highly self-selected sample what they thought about the testing system will give rise to responses of unknown generalisability. Those who participated are plausibly those who wanted to be tested regularly in the first place, and were quick to take up the opportunity for something more accurate and less invasive than LFTs. More emphasis could be put on these severe limitations. More focus could have been directed in real time at uncovering the reasons for people not participating. The post hoc probe using the student panel is a start. As the final sentence suggests, further study is needed. Questions were asked about the perceived effect of vaccination status on behaviour, self-perceptions of 'being protected', participation and on response to the test result, but the answers are not recorded. Nor are those to the questions on personal experience of prior Covid infection, although a majority of survey participants are recorded as being moderately or very concerned about catching covid before taking part in TestEd. If neither vaccination status nor past covid infection appeared to have any relevant effects, that should be stated. There is a suggestion from a reported survey response that staff and student perceptions of their risk are likely to be different: the former may spend much less time in the company of others, since more likely to have their own work space. The practicalities of self-isolation following a positive test are also likely to be very different for students compared with staff. Again if there really was no difference between the two groups, at least in respect of self-reported compliance, that is worth stating.
---

VERSION 1 – AUTHOR RESPONSE

Reviewer 1's comments	Response
Could the authors provide any details about what percentage of staff and students registered and consented to participate, and if known, what percentage this is of those who regularly attended campus?	During the study period (when Covid-19 public health measures were in place) most staff and students were working from home. We do not know what proportion of all students or staff would have been eligible to register, because registration was contingent on visiting campus and data on this was not kept by the University except to monitor building occupancy, for example. It is also worth noting that some

	students and staff were not even in Edinburgh during the pandemic and instead studying or working remotely from elsewhere. We mentioned in the limitations section but we have now expanded this to make it much clearer (page 13). We have also clarified on page 4 about eligibility being for those on campus.
Were the 170 positive tests based purely on university PCR tests? Or is this those that were also confirmed through NHS PCR tests?	The 170 is based on the University PCR tests. However, all of these participants were advised to seek a confirmatory NHS PCR test and had to show proof of that to the TestEd project manager (via email/text) in order to be eligible for the voucher to support self-isolation. This point is already included in the first paragraph of page 4.
Were you able to obtain data on the number of participants who also sought a PCR, and if so, how reliable were the university tests for detecting infection?	All the survey and interview participants who tested positive with TestEd had this confirmed with an NHS test. A separate paper is planned from other members of the TestEd team responsible for the lab elements on sensitivity and specificity, which isn't the focus of this paper.
The description of the qualitative analysis could be more detailed – for example, how did the authors move from codes to themes? Were any approaches used to enhance rigor/credibility of the analysis?	We have provided additional detail on the process for thematic analysis and review of themes on page 5.
Ethical approval should be mentioned.	Apologies this was an oversight. Details of ethical approval now included at the end of the 'Design' section (page 4)
That testing did not (really) change behavior is an interesting finding. I would be interested to hear the authors' views about this, why this may have been (e.g., what did behavior before testing look like) and whether or not behavior (within the guidance) should have changed (which is arguably/debatably the point of testing).	A paragraph on this is already included in the discussion on page 13 and it is similar to other studies of University testing programmes. We have added a sentence to expand this.
Reviewer 2's comments	Response
The main limitation of this paper is the uncertainty about representativeness of the participants ...Asking a small portion of an already highly self-selected sample what they thought about the testing system will give rise to responses of unknown generalisability. Those who participated are plausibly those who wanted to be tested regularly in the first place, and were quick to take	In response to reviewer 1 we have expanded the discussion to emphasise this limitation (page 13). We have also added to the paragraph on limited changes in behaviour among those who participated and that they are likely to have been a group that was more engaged than others and keen to follow Covid-19 guidelines (see above to reviewer one, 5 th

up the opportunity for something more accurate and less invasive than LFTs. More emphasis could be put on these severe limitations.	para page 13)
More focus could have been directed in real time at uncovering the reasons for people not participating. The post hoc probe using the student panel is a start. As the final sentence suggests, further study is needed.	We agree, although the focus of the TestEd study overall was to examine the uptake and outcomes of the programme itself and the views of those who participated, rather than those who did not. We've acknowledged this in the limitations section and made clear that we were limited to asking an existing student panel why they didn't participate if they had not accessed TestEd (2nd paragraph page 14)
Questions were asked about the perceived effect of vaccination status on behaviour, self-perceptions of 'being protected', participation and on response to the test result, but the answers are not recorded. Nor are those to the questions on personal experience of prior Covid infection, although a majority of survey participants are recorded as being moderately or very concerned about catching covid before taking part in TestEd. If neither vaccination status nor past covid infection appeared to have any relevant effects, that should be stated.	Perceptions of protection from vaccination and previous experiences of Covid were both discussed by interview participants in relation to testing motivations. We have reported this on P7.
There is a suggestion from a reported survey response that staff and student perceptions of their risk are likely to be different: the former may spend much less time in the company of others, since more likely to have their own work space. The practicalities of self-isolation following a positive test are also likely to be very different for students compared with staff. Again if there really was no difference between the two groups, at least in respect of self-reported compliance, that is worth stating.	Challenges in self-isolation were mentioned by both students and staff with particular detail elicited in interviews. The interviews were not intended to be representative so we need to be cautious about suggesting this applies to either all staff or students. However we have now added to the second paragraph on page 12 that students highlighted particular difficulties.